# Daylight Photodynamic Therapy for Actinic Keratosis and Field Cancerization: A Narrative Review

**DOI:** 10.3390/cancers17061050

**Published:** 2025-03-20

**Authors:** Elena Sotiriou, Dimitra Kiritsi, Nikolaos Chaitidis, Michael Arabatzis, Aimilios Lallas, Efstratios Vakirlis

**Affiliations:** 1First Department of Dermatology and Venereology, Faculty of Medicine, Aristotle University of Thessaloniki, 54124 Thessaloniki, Greece; elenasotiriou@yahoo.gr (E.S.); nchaitidauth@gmail.com (N.C.); marabatzis@auth.gr (M.A.); emlallas@gmail.com (A.L.); svakirlis@hotmail.com (E.V.); 2Department of Dermatology, Medical Center, Faculty of Medicine, University of Freiburg, 79085 Freiburg, Germany

**Keywords:** actinic keratosis, sun-damaged skin, daylight photodynamic therapy, field cancerization

## Abstract

For patients with actinic keratosis (AK) and chronic sun damage, it is crucial to treat both AK and field cancerization to prevent skin cancer. While most AKs do not progress to cutaneous squamous cell carcinoma (cSCC), the majority of cSCC cases arise from AKs. Photodynamic therapy (PDT) effectively targets both non-hyperkeratotic AK lesions and surrounding sun-damaged skin, reducing the risk of new AKs. Though PDT is proven to be safe and effective, it is underused due to the need for office-based equipment and associated pain. Daylight PDT, which uses natural sunlight for activation, could make PDT more accessible in everyday dermatological care.

## 1. Introduction

Actinic keratoses (AKs), also known as solar keratoses, are rough, scaly lesions that appear as macules, papules, or plaques. AKs represent premalignant growths. They develop due to the abnormal proliferation of atypical keratinocytes within the epidermis, triggered by long-term exposure to ultraviolet (UV) radiation. Field cancerization in AKs refers to the phenomenon where the skin in sun-exposed areas undergoes widespread genetic and cellular alterations due to chronic UV radiation exposure. This process creates a “field” of cells that are genetically predisposed that may progress to AKs. These lesions may be at varying stages of malignant transformation, with the underlying skin often harboring subclinical abnormalities that increase the risk of developing AKs and subsequent squamous cell carcinoma, if left untreated [1,2].

For patients suffering from AKs and clinical evidence of chronic sun damage, achieving both actinic keratosis eradication and treating field cancerization is essential [3,4]. AKs should not be underestimated. While the majority do not progress to cutaneous squamous cell carcinoma (cSCC), most cases of cSCC develop from pre-existing AKs. Importantly, there is no reliable way to distinguish which AKs will progress to cSCC and which will spontaneously regress or remain unchanged [5,6]. Actinic keratoses and field cancerization are chronic conditions that require multiple treatment cycles and long-term patient follow-up [4,5,6].

Various effective treatments are available for AKs, including lesion-directed treatments (cryosurgery, surgery, dermabrasion, ablative laser) and field-directed treatments (fluorouracil, imiquimod, tirbanibulin, diclofenac, photodynamic therapy, retinoids, chemical peels, laser resurfacing) (Table 1). Treatment choice depends on factors such as the number and distribution of lesions, lesion characteristics, patient preference, tolerance for side effects, and treatment availability and cost. For few, isolated lesions, and hyperkeratotic lesions, liquid nitrogen cryosurgery is preferred due to its quick procedure, cost-effectiveness, and good cosmetic results. For hypertrophic or hyperkeratotic lesions, cryosurgery with two freeze–thaw cycles is commonly suggested. For multiple thin lesions or field cancerization, 5 fluorouracil (5-FU), imiquimod or photodynamic therapy are recommended. In patients with multiple lesions, including hypertrophic ones, a sequential therapy approach is recommended. This typically involves using cryosurgery to treat the hypertrophic lesions, followed by field treatments (such as 5-FU, imiquimod or PDT) for the surrounding cancerized field. This combination approach allows for targeted treatment of more extensive or resistant lesions while addressing field cancerization effectively [4,7,8,9].

PDT is particularly effective in treating both non-hyperkeratotic AK lesions and the surrounding sun-damaged skin containing precancerous cells. This dual action reduces the risk of new AK formation, making PDT a valuable option for comprehensive skin cancer prevention in chronically sun-exposed individuals. The conventional PDT (cPDT) has already been proven a safe and effective method to treat actinic keratosis and field cancerization [6,7,10,11,12]. However, in the real world, PDT has not reached its maximum potential use, possibly due to the need of an office-based light source equipment and the associated peri-procedural pain (Table 2) [7,13,14]. Daylight PDT (dPDT), which utilizes natural sunlight to activate the photosensitizer and eradicate precancerous cells, offers a promising alternative that may help integrate PDT more effectively into routine dermatological care [15].

**Table 1 cancers-17-01050-t001:** Lesion-directed and field-directed treatments for actinic keratosis and field cancerization [1,4,7,8,12,13,15,16,17,18,19].

Treatment Type	Treatment Modality	Mechanism of Action	Application	Indication
Lesion-Directed Treatments	Cryotherapy	Induces cellular necrosis through rapid freezing of the lesion	Liquid nitrogen applied directly to lesion	Isolated, well-demarcated AK lesions
	Surgical Excision	Complete excision of the lesion with scalpel or surgical instrument	Direct surgical removal of lesion	Thick, hypertrophic, or clinically suspicious AK
	Curettage and Electrodessication	Physical scraping followed by electrocautery to eliminate abnormal keratinocytes	Curette scraping followed by electrocautery	Hypertrophic or hyperkeratotic AK lesions
	Ablative Laser Therapy	Utilizes ablative laser to vaporize and remove AK lesions	CO_2_ or Er:YAG laser applied directly	Localized AK lesions or cosmetic concerns
Field-Directed Treatments	5-FU	Inhibits DNA synthesis and induces apoptosis in abnormal keratinocytes	Topical cream applied to affected areas	Multiple AK lesions or areas with field cancerization
	Imiquimod	Modulates the immune system to enhance the immune-mediated clearance of AK lesions	Topical cream applied 2–3 times per week	Multiple AK lesions or field cancerization
	Diclofenac (NSAID Gel)	Inhibits COX-2, leading to apoptosis of AK cells	Topical gel applied twice daily	Mild to moderate AK lesions
	Tirbanibulin	Inhibits microtubule polymerization, inducing selective apoptosis of AK cells	Topical ointment applied once daily for 5 consecutive days	Mild AK lesions on the face and scalp
	cPDT	Photosensitization of keratinocytes through a photosensitizing agent, followed by light exposure to generate ROS.	Topical application of MAL or ALA followed by light activation	Multiple AK lesions and field cancerization
	dPDT	Utilizes natural sunlight to activate the photosensitizing agent, inducing apoptosis through ROS generation	Topical application of MAL or ALA followed by 1–2 h of sun exposure	Extensive AK lesions or field cancerization with reduced pain

AK: actinic keratosis; ALA: 5-aminolevulinic acid; COX-2: cyclooxygenase-2; cPDT: conventional photodynamic; dPDT: daylight photodynamic therapy; MAL: methyl aminolevulinate; NSAID: non-steroidal anti-inflammatory drug; ROS: reactive oxygen species; 5-FU: 5-Fluorouracil.

**Table 2 cancers-17-01050-t002:** Adverse events associated with conventional photodynamic therapy (cPDT) and their frequency [14,20,21,22,23,24,25,26,27,28].

Adverse Events	Description	Frequency
Pain, erythema, pruritus, crusting/scaling	Severe pain may occur in a significant proportion of patients, redness, scabbing	Common
Dyschromia, photosensitivity reaction	Skin darkening or lightening, delayed sunburn-like reaction	Uncommon
Infection, erosion, ulceration, edema	Secondary bacterial infection, skin breakdown, swelling	Rare

## 2. Methodology

We conducted a literature review to compare dPDT with other treatment modalities for actinic keratosis (AK) and/or field cancerization. Our search strategy included the following Boolean search terms: (solar OR actinic OR field) AND daylight AND phototherapy. We searched the following databases: PubMed/MEDLINE, Cochrane Library, Scopus, Web of Science, ClinicalTrials.gov, and Google Scholar. Additionally, we manually screened the reference lists of all eligible studies to identify relevant publications. Only studies published in English and Greek were included in our review.

Studies were eligible for inclusion if they reported outcomes comparing dPDT with other field-directed or lesion-directed treatment modalities for AK. Studies involving only dPDT-treated patients and case report studies were excluded. Moreover, studies reporting on the safety and efficacy of dPDT on other skin conditions were also excluded. Data extraction was conducted independently by two investigators (NC, MA), with any discrepancies resolved through consultation with a third investigator (DK). Due to the narrative nature of our review, a targeted quality assessment was performed solely to provide an overview of the included studies’ methodological rigor. The quality of the included studies was assessed using appropriate tools based on study design. For randomized controlled trials, the Cochrane Risk of Bias 2 (RoB 2) tool was used to evaluate bias across key domains [29]. For non-randomized studies, the Risk of Bias in Non-Randomized Studies of Interventions (ROBINS-I) tool was applied to assess potential biases [30].

## 3. Review of the dPDT Procedure

Initially, patients should apply a chemical sunscreen with a sun protection factor (SPF) greater than 20, free of physical filters, to all sun-exposed areas, including the treatment site. Chemical sunscreens absorb UV radiation while permitting visible light to pass through. This ensures skin protection during sun exposure while still allowing visible light to activate the photosensitizing agent [15]. Pretreatment with curettage, keratolytics, 5-fluorouracil (5-FU), or laser may enhance the penetration of the photosensitizing agent, potentially improving treatment efficacy [11,15,31,32].

After allowing sufficient time for the sunscreen to be absorbed, patients should proceed with applying the photosensitizing agent, typically methyl aminolevulinate (MAL) or 5-aminolevulinic acid (ALA), to the affected areas [15]. Occlusion after photosensitizer application is optional but may enhance efficacy without increasing adverse event rates [15,33]. In Greece, MAL is the only commercially available photosensitizing agent.

Following application of the photosensitizer, the patient should undergo two hours of natural sunlight exposure—whether sunny or cloudy—to allow for the gradual activation of the photosensitizer. This process selectively targets and destroys precancerous keratinocytes in AKs and subclinical lesions within the cancerized field. The photosensitizing agent penetrates keratinocytes, where it is metabolized into protoporphyrin IX (PpIX), a photosensitive molecule. Regarding the photosensitizer distribution, limited data have shown that ALA-derived PpIX fluorescence in skin increases post-application, plateauing at 4–14 h depending on the concentration, dosage, and application time of the photosensitizer [34]. Upon exposure to natural daylight, absorbed light energy leads to light-induced degradation of PpIX (i.e., photobleaching) and the generation of reactive oxygen species (ROS) selectively within the premalignant cells and the endothelial cells of the tumor vasculature. These ROS critically damage cell membranes, mitochondria, and other vital cell organelles, ultimately triggering apoptosis and/or necrosis of the affected cells. PDT may also involve an immunological mechanism, activating innate and adaptive immune cells through antigen presentation of the destroyed premalignant cell [35,36,37]. However, its role remains unclear, as a clinical immunohistochemical study on PDT for basal cell carcinoma found increased neutrophil infiltration and E-selectin expression post-treatment, but also a significant loss of Langerhans cells (specialized skin antigen-presenting cells), suggesting that PDT-induced immunosuppression may weaken antitumor immunity [38].

The gradual activation of the photosensitizing agent during daylight exposure is likely the main reason why this procedure causes significantly less pain compared to conventional PDT (cPDT) (Table 3). To minimize discomfort, it is crucial that patients expose their skin to sunlight within 30 min after photosensitizer application. Delayed exposure (beyond 30 min after photosensitizer application) can lead to excessive PpIX accumulation, which may increase pain levels [15,39].

Easy-to-apply smartphone-based fluorescence imaging for quantitative PpIX measurement opens new possibilities for personalized PDT dosimetry and improved clinical study design, as reduced PpIX accumulation and photobleaching correlate with lower treatment efficacy [40,41,42].

After two hours of daylight exposure, the photosensitizer must be removed, and the treated area should be protected from sunlight for the remainder of the day to prevent excessive inflammation. Between 24 to 48 h after sun exposure, erythema resembling sunburn typically appears, lasting for a few days (Table 4). During this phase, the daily use of mild cleansers and application of emollient creams or specialized dermocosmetics is recommended to soothe discomfort and aid in healing, particularly in cases of erosion or crusting formation [15,43].

It is crucial to recognize that when hyperkeratotic AKs are present, lesion-directed ablation should be prioritized to maximize treatment effectiveness. Subsequently, dPDT can be applied to non-hyperkeratotic AKs and the surrounding cancerized field, as dPDT tends to be less effective in treating hyperkeratotic lesions [15].

**Table 3 cancers-17-01050-t003:** Comparison of daylight photodynamic therapy (dPDT) vs. conventional photodynamic therapy (cPDT) for actinic keratosis and field cancerization [1,4,7,8,13,15,18,21,22,44].

Aspect	Daylight PDT (dPDT)	Conventional PDT (cPDT)
Efficacy	Effective for non-hyperkeratotic AK and field cancerization.	Effective for non-hyperkeratotic AK and field cancerization.
Pain and Tolerability	Much less painful due to gradual activation of photosensitizer.	Often painful due to rapid activation with intense light.
Cosmetic Outcome	Excellent, with minimal inflammation and scarring.	Also good, but potential for more post-treatment erythema and irritation.
Convenience	No need for artificial light sources; can be performed outdoors.	Requires a specialized light source and clinical setup.
Treatment Setting	Can be performed outside or indoors near windows.	Requires a clinical setting with trained personnel.
Weather Dependence	Dependent on sufficient daylight (not ideal for cloudy/rainy days).	Independent of weather conditions.
Cost and Equipment	More cost-effective (no expensive light source required).	Higher costs due to specialized light equipment and clinical visits.
Treatment Time	Longer exposure (2 h outdoors), but shorter clinic time.	Shorter exposure (7–10 min per lesion) but longer clinic visits.
Patient Compliance	Easier for patients due to minimal pain and fewer clinic visits.	Compliance may be lower due to pain and frequent clinic visits.
Adverse Effects	Milder side effects (low pain, mild erythema, some scaling).	More erythema, swelling, crusting, and pain post-treatment.
Recurrences	Comparable efficacy in mild-to-moderate AK, good for field cancerization.	Potentially better for thicker AKs but with higher local inflammation.

**Table 4 cancers-17-01050-t004:** Adverse events associated with daylight photodynamic therapy (dPDT) and their frequency [14,20,21,22,23,24,25].

Adverse Events	Description	Frequency
Pain, erythema, pruritus, crusting/scaling	Mild to moderate burning or itching, redness, scabbing	Common
Dyschromia, photosensitivity reaction	Skin darkening or lightening, delayed sunburn-like reaction	Uncommon
Infection, erosion, ulceration	Secondary bacterial infection, skin breakdown	Rare

## 4. Review of Current Evidence

### 4.1. Comparison of dPDT with cPDT

Ten randomized trials have compared the efficacy and safety of dPDT with cPDT. Our literature search did not identify any studies in Greek that were eligible for inclusion in our review. Based on the quality assessment, the overall methodological quality of the included studies was deemed to be low. Eight studies focused on natural dPDT in comparison with cPDT, while two investigated artificial dPDT in comparison to cPDT [14,20,21,22,23,24,39,44,45,46,47,48,49]. All but two trials reported similar short-term efficacy between the two methods in treating existing AKs [20,22,23,24,39,44,46,47,49]. Exceptions included Fargnoli et al., who reported significantly higher clearance rates at 3 months for cPDT when treating thicker AKs (grades II and III), whereas no differences were noted between the two treatment options for grade I AKs [21]. The same study also found significantly better AK clearence rates for the cPDT group at 12 months, but no difference at the 12-month recurrence rates between the two treatment groups [45]. Additionally, Wiegel et al., who compared cPDT with ultra-low artificial dPDT, found significantly higher response rates in patients treated with conventional PDT [23].

Regarding adverse events, all but one study reported significantly less pain in patients receiving dPDT, while one study reported similar post-procedural pain levels among the participants [47]. It is worth noting that all studies included a small sample size (≤100 patients per study) [14,20,21,22,23,24,39,44,46,49].

### 4.2. Comparison of dPDT with Other Treatments Modalities and Combinations of Treatments

Unfortunately, only a handful of studies compared dPDT with other lesion-directed or field-directed treatment modalities. The quality assessment classified the studies’ methodological quality as low. A pilot study by Galimberti et al. evaluated dPDT with 16% MAL cream versus 5-FU cream for AK on the face and scalp in five male participants. After three months, both treatments were effective (80% complete response for dPDT and 93% for 5-FU), but dPDT had fewer side effects, quicker recovery, and higher patient preference. The findings suggest that dPDT is an effective and patient-friendly alternative to 5-FU for managing AKs [50].

A randomized trial evaluated repetitive dPDT versus cryosurgery for AKs in 58 patients with more than five AKs on sun-damaged facial skin. Over two years, patients receiving dPDT developed fewer new AKs (mean 7.7 lesions) compared to cryosurgery (mean 10.2 lesions), with the difference nearing statistical significance (*p* = 0.18). The authors stated that the lower rate of new AKs in the dPDT group may have reflected the concomitant treatment of field cancerization, though this remains a hypothesis. They also found that dPDT also significantly improved signs of photoaging and caused less pain and fewer side effects than cryosurgery [42].

However, it is not necessary to choose only one treatment option among the plenty. Combining treatments appears to improve efficacy of AK and field cancerization treatment, as shown in the randomized trial by Wiegel et al. Their study demonstrated that sequential 4% 5-FU followed by dPDT resulted in a higher clearance rate at three months (87% vs. 74%) compared to dPDT alone, particularly for grade II AKs (79% vs. 55%) [11]. At the six-month follow-up, the clearance rate for the combination treatment was 84%, compared to 69% for dPDT alone. At twelve months, the clearance rate remained higher for the combination (79%) compared to dPDT alone (70%). The combination therapy was especially effective for moderately thick AKs. Recurrence rates were also lower with 5-FU + DL-PDT at both the six months (10%) and twelve months (15%) follow-ups, compared to dPDT alone (20%). Furthermore, the number of new lesions was fewer in the combination group. Patient satisfaction was higher with 5-FU + dPDT (85%) compared to dPDT alone (71%), due to the improved efficacy, despite the increased erythema and pain associated with the combination treatment [11,51].

A randomized intra-individual study conducted by Nissen et al. found that pretreatment with 5-FU enhanced the efficacy of dPDT for acral AKs in 24 patients. The combination therapy had a significantly higher lesion response rate (62.7%) compared to dPDT alone (51.8%) at three-month follow-up, with similar pain and erythema in both groups [52]. Another randomized intra-individual study by Piaserico et al. which compared topical calcitriol + dPDT versus placebo and dPDT) for treating acral AKs in 42 patients found that combination treatment showed a higher lesion response rate for grouped grade II/III AKs; local skin reactions were more frequent with combination treatment [53].

A single-blinded, randomized, intra-individual study by Lindholm et al. investigated whether fractional laser pre-treatment enhances the efficacy of dPDT for AKs of all grades and compared the outcomes of artificial and natural dPDT. At the 6-month follow-up, fractional laser-mediated dPDT achieved significantly higher complete clearance (50.0% vs. 30.3%, *p* = 0.04), partial clearance (78.6% vs. 50.0%, *p* < 0.01), and lesion-specific clearance (86.2% vs. 70.2%, *p* < 0.01) compared to dPDT alone. However, no significant differences were observed between artificial and natural dPDT or between grade I and grade II–III lesions. The authors concluded that fractional laser pre-treatment significantly enhances the efficacy of both artificial and natural dPDT, making it a suitable approach for treating AKs of all grades [32].

### 4.3. Significance of Field Cancerization Treatment

Treating field cancerization in AK is significant because it addresses both visible lesions and the surrounding skin that may harbor subclinical damage, thereby reducing the risk of recurrence and potential progression to cSCC. Field-directed therapies, such as PDT, 5-FU, and imiquimod, have demonstrated efficacy in clearing AK lesions and treating the broader area of damaged skin. For instance, clinical studies have reported lesion clearance rates ranging from 81% to 91% for PDT. Additionally, field-directed treatments may reduce the risk of AK recurrence and potentially lower the risk of developing cSCC [4,8,9,13,16].

By targeting the entire field of cancerization, these treatments not only manage existing lesions but also prevent the emergence of new ones, leading to improved long-term skin health and patient outcomes.

## 5. Discussion

Daylight PDT is an effective and safe treatment for non-hyperkeratotic AKs and field cancerization. Its advantages include absence of specialized equipment, limited or even single-session applications and well-tolerated adverse effects, making it highly appealing to both physicians and patients. However, its drawbacks include lower efficacy in treating hyperkeratotic AKs and challenges in regions with limited daylight or adverse weather conditions for a significant portion of the year [11,14,15,20,21,22,23,24,39,44,45,47,48,49].

Since AK and field cancerization are chronic conditions, patient monitoring and close follow-up after dPDT sessions is crucial [15]. Daylight PDT is a key option for managing both non-hyperkeratotic AKs and field cancerization, particularly as alternative field treatments, such as 5-FU and imiquimod, are linked to less tolerable side effects, including pain, severe erythema, and crusting, which can interfere with patient adherence [10,54,55]. Additionally, treatments like cPDT are associated with significant pain and necessitate specialized equipment and trained personnel, further limiting their accessibility in real-world circumstances [15].

The reduced efficacy of dPDT in hyperkeratotic AKs can be mitigated by preceding the therapy with lesion-targeted treatments, such as cryosurgery, to address hyperkeratotic lesions before initiating dPDT [15]. We propose a practical algorithm for the treatment of AKs and field cancerization (Figure 1). Based on this algorithm, the treating dermatologist should assess the extent of damage on chronically sun-exposed areas, including the face, scalp, neck, dorsal hands, and forearms. If AKs are present, initial lesion-directed treatment with cryotherapy is recommended, particularly for hyperkeratotic AKs, as these lesions may be less responsive to subsequent field-directed therapies. Cryotherapy remains a widely used and effective method, even for hyperkeratotic AKs. Following an adequate healing period—typically a few days up to 2–3 weeks, potentially expedited with adjunctive dermocosmetic products—patients should undergo field therapy with dPDT. This approach targets subclinical actinic damage and any residual non-hyperkeratotic AKs, optimizing overall treatment efficacy. Given the chronic nature of AKs and the potential for malignant transformation, long-term surveillance is essential. Regular follow-up ensures treatment success, facilitates early detection of recurrent or new lesions, and allows for timely intervention with additional preventive and therapeutic strategies, including photoprotection.

Additionally, while weather conditions may pose challenges, dPDT has demonstrated high clearance rates even under unfavorable conditions [23]. In such scenarios, artificial light (i.e., indoor) dPDT variants offer a practical alternative, particularly during the darker months in northern regions [24,47].

Evidence from real-world studies supports the findings from the randomized trials that were reviewed in previous sections. Phillipp-Dormston et al. evaluated the effectiveness and practicality of MAL-based artificial dPDT for grade I and II AKs under real-world conditions in 224 patients. After three months, AK lesions in the treated focus area were reduced by 71% (*p* < 0.001), with most patients (93.3%) reporting no or mild pain during the procedure. The treatment also significantly decreased the Actinic Keratosis Area and Severity Index (AKASI) and received high satisfaction ratings from both patients (80.0%) and investigators (82.8%). These results highlight dPDT as an effective and well-tolerated option for managing AK on the face and scalp [57].

An Australian observational study evaluated the use of dPDT for treating mild to moderate AKs on the face and scalp. After a single treatment, nearly half of the patients (46.8%) required no further treatment, with high satisfaction rates from both patients (79.7%) and physicians (83.3%). The procedure was well-tolerated, with most patients reporting minimal to no pain (74.1%). Adverse events were mild, primarily erythema (44.4%). Overall, dPDT proved to be an effective, convenient, and well-tolerated option for managing AK [58].

Patients’ preferences regarding different treatment modalities should also be considered. In a study involving 100 patients, dPDT demonstrated higher overall satisfaction compared to other field-directed treatments, including 5-FU, imiquimod, diclofenac, and cPDT. The authors suggested that the shorter treatment regimen may contribute to the higher satisfaction levels observed in patients undergoing dPDT [59].

Regarding cost-benefit analysis, the literature is lacking comparison of dPDTs with other treatment modalities. A Finnish randomized trial evaluated cost-effectiveness of dPDT and artificial dPDT for treating AKs. The study included 70 patients. Daylight PDT was found to be less costly (EUR 132 vs. EUR 170) but less effective, with a lower complete response rate. The incremental cost-effectiveness ratio indicated that dPDT offered less value for money due to its reduced effectiveness despite lower costs [60]. A systematic review of pharmacoeconomic studies regarding AK treatment by Vale at al. concluded that PDT, ingenol mebutate, and 5-FU were the most cost-effective treatments for AK, with PDT offering superior cosmetic outcomes and better patient tolerance. However, according to the authors, the review was limited by the absence of direct comparisons between different treatment modalities and inconsistencies across the studies, with conflicts of interest in many of the included studies also being a factor to consider [56].

## 6. Limitations of the Review

Our review carries the inherent limitations of narrative reviews. Moreover, studies on the treatment of AK, including those on dPDT, are often of low quality due to factors such as small sample sizes, inconsistent study designs, and short follow-up periods. Additionally, inadequate reporting of outcomes, biases, and confounding factors weaken the evidence. Variations in treatment modalities and potential conflicts of interest further complicate the comparison and evaluation of treatments.

## 7. Suggestions for Future Research

Future research in dPDT for AKs and field cancerization should focus on large-scale studies to evaluate long-term efficacy and recurrence rates, particularly for squamous cell carcinoma. Studies optimizing sunlight exposure duration, intensity, and photosensitizer formulations could improve treatment outcomes. Additionally, comparative effectiveness studies between dPDT and other therapies, such as cryotherapy or 5-FU, would help determine the most cost-effective and patient-preferred treatment. Research into patient-specific factors like skin type and ultraviolet exposure could further personalize treatment approaches. Lastly, exploring combination therapies with dPDT could enhance its therapeutic efficacy and reduce recurrence.

## 8. Conclusions

In conclusion, dPDT is an effective, well-tolerated, and convenient treatment for AKs and field cancerization, providing high satisfaction rates for both patients and physicians. While its efficacy may be diminished in hyperkeratotic lesions and areas with severely limited daylight, combining dPDT with lesion-targeted treatments and utilizing indoor alternatives can address these challenges. The reduced peri-procedural pain and fewer topical adverse effects offer a significant advantage over other field treatments, such as 5-FU, imiquimod, and cPDT, rendering dPDT a highly practical option in real-world clinical settings.

## Figures and Tables

**Figure 1 cancers-17-01050-f001:**
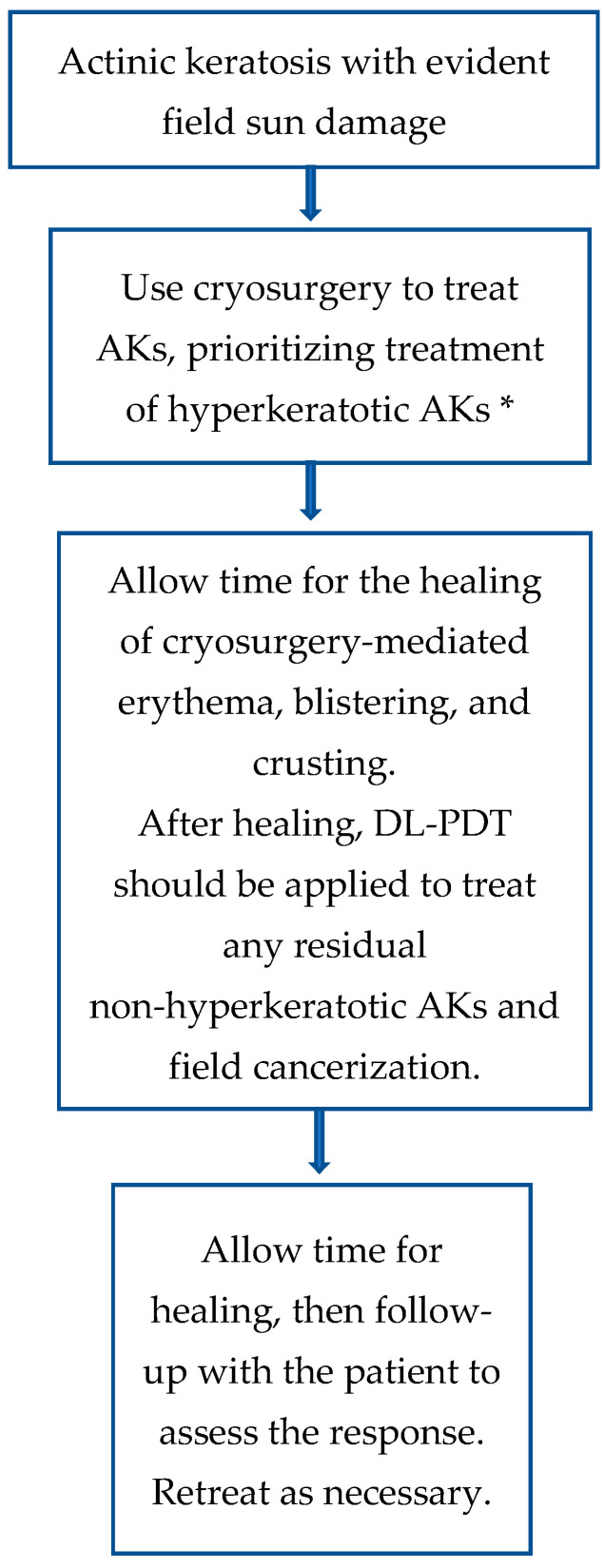
Practical algorithm to treat AK and field cancerization [1,2,7,8,12,13,15,17,18,56]. * It is recommended to use mild cleansers, emollients, and specialized dermocosmetics to facilitate post-procedural healing. AK: actinic keratosis; dPDT: daylight photodynamic therapy.

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
