# Peer review of "Daylight Photodynamic Therapy for Actinic Keratosis and Field Cancerization: A Narrative Review"

_cancers, 2025, doi:10.3390/cancers17061050_

Round 1
Reviewer 1 Report
Comments and Suggestions for Authors
The manuscript submitted for review was prepared carefully and professionally. My only remark concerns the technical side of this manuscript. It is suggested that this work be supplemented with additional diagrams/figures.
Author Response
C1: The manuscript submitted for review was prepared carefully and professionally. My only remark concerns the technical side of this manuscript. It is suggested that this work be supplemented with additional diagrams/figures.
A: Thank you for your comment. We included two more tables as suggested (Table 3 and Table 4).
Reviewer 2 Report
Comments and Suggestions for Authors
The manuscript presents a comprehensive analysis of the published literature on photodynamic daylight therapy (dPDT) as a treatment for actinic keratosis (AK). It clearly presents the aims and rationale for the study and emphasises the need for more accessible, well-tolerated treatments for AK, particularly in contrast to conventional PDT (cPDT) and other field-directed therapies such as 5-FU and imiquimod.
The article includes a methodology section describing the authors' approach to the literature search. It contains the search criteria, the sources used, but no explicit mention on the inclusion and exclusion criteria, the data extraction methods and the quality assessment of the included studies is included in this section.
The tables provide comparative insights, particularly in differentiating dPDT from cPDT and other treatment options. However, the practical treatment algorithm mentioned in the discussion is not fully explained in the text, which may increase the usefulness of the manuscript. A cost-benefit analysis, either as a table or as an additional discussion point, would add relevance to the study.
While the manuscript emphasises the benefits of dPDT in terms of patient adherence and reduction of side effects, long-term data comparing dPDT with cPDT, 5-FU and imiquimod in terms of recurrence of AK and reduction of SCC risk are lacking. Other strengths of dPDT, such as its accessibility, tolerability and convenience, as well as the potential of combination treatments to increase its efficacy, are also cited. While some limitations are acknowledged — such as reduced efficacy in hyperkeratotic AKs and dependence on sunlight availability — the manuscript could include a more critical discussion of other challenges. These include the variability of dPDT protocols in the selected studies, patient-specific factors that influence treatment response, and the lack of large-scale randomised trials with long-term follow-up.
The structure and flow of the manuscript are generally logical, but some reorganisation could improve readability. The discussion summarises several aspects— - the effectiveness of PPDT, its limitations, evidence from practise and future research— - in one dense section. Splitting it into separate sections would improve clarity. In addition, moving some details from the discussion to the methodology (e.g. the selection of studies) would lead to a better structure of the report.
Author Response
C1: The manuscript presents a comprehensive analysis of the published literature on photodynamic daylight therapy (dPDT) as a treatment for actinic keratosis (AK). It clearly presents the aims and rationale for the study and emphasizes the need for more accessible, well-tolerated treatments for AK, particularly in contrast to conventional PDT (cPDT) and other field-directed therapies, such as 5-FU and imiquimod.
The article includes a methodology section describing the authors' approach to the literature search. It contains the search criteria, the sources used, but no explicit mention on the inclusion and exclusion criteria, the data extraction methods and the quality assessment of the included studies is included in this section.
A: Thank you for your comment. We revised the Methodology section in order to ensure it contains all the relevant information as instructed.
C2: The tables provide comparative insights, particularly in differentiating dPDT from cPDT and other treatment options. However, the practical treatment algorithm mentioned in the discussion is not fully explained in the text, which may increase the usefulness of the manuscript.
A: We now provide an explanation of the practical treatment algorithm in the discussion section as instructed.
C3: A cost-benefit analysis, either as a table or as an additional discussion point, would add relevance to the study.
A: We have added an additional point in the Discussion section regarding the cost-benefit of treating actinic keratosis (AK) with daylight photodynamic therapy (dPDT), based on the available literature.
C4: While the manuscript emphasizes the benefits of dPDT in terms of patient adherence and reduction of side effects, long-term data comparing dPDT with cPDT, 5-FU and imiquimod in terms of recurrence of AK and reduction of SCC risk are lacking. Other strengths of dPDT, such as its accessibility, tolerability and convenience, as well as the potential of combination treatments to increase its efficacy, are also cited. While some limitations are acknowledged — such as reduced efficacy in hyperkeratotic AKs and dependence on sunlight availability — the manuscript could include a more critical discussion of other challenges. These include the variability of dPDT protocols in the selected studies, patient-specific factors that influence treatment response, and the lack of large-scale randomized trials with long-term follow-up.
A: We have revised the Discussion section to offer a more thorough and critical appraisal of studies reporting on treatments for AK, including dPDT.
C5: The structure and flow of the manuscript are generally logical, but some reorganisation could improve readability. The discussion summarises several aspects— - the effectiveness of PPDT, its limitations, evidence from practise and future research— - in one dense section. Splitting it into separate sections would improve clarity. In addition, moving some details from the discussion to the methodology (e.g. the selection of studies) would lead to a better structure of the report.
A: Thank you the reviewer for the comments, helping us improve the manuscript’s readibility. The manuscript has been reorganized to separate the aspects of limitations, and suggestions for future research into distinct sections for improved clarity. Additionally, we have moved relevant details, such as the selection of studies, from the discussion to the methodology section, enhancing the overall structure of the manuscript."
Reviewer 3 Report
Comments and Suggestions for Authors
1. The text of the article does not contain enough references to the literature to support the statements made. The tables are completely devoid of references.
2. The text of the article contains many unsubstantiated conclusions that require the inclusion of references.
3. In the Methodology section, the authors indicate that the review included only studies published in Greek and English. However, the review does not contain references to articles in Greek.
4. Please describe in more detail the side effects associated with the use of both cPDT and dPDT? What are their main manifestations and how often do they occur?
5. What is the underlying mechanism, membrane damage or mitochondrial dysfunction leads to apoptosis? Explain what is meant by oxidative stress.
6. What is the dynamics of photosensitizer accumulation from the moment of application of ointment with PS to the formation of a sufficient amount of PpIX in actinic keratosis cells for effective treatment?
7. The text of the article does not present an analysis of modern methods for monitoring the accumulation and photobleaching of PS in tissues.
8. How many dPDT sessions are needed to achieve full therapeutic effect?
9. What is the role of Langerhans cells in the dPDT Actinic Keratosis process? Do they influence the effectiveness of therapy or the immune response?
10. How painless is dPDT? The authors mention that dPDT is virtually painless. However, there are enough studies that mention the occurrence of pain during 5-ALA dPDT.
11. The authors mention that patients should go out into daylight within 30 minutes of applying PS. Please clarify whether it is recommended to start exposure to daylight immediately after applying the ointment or whether it is necessary to wait a certain amount of time. This interesting point makes sense to expand on in order to remove the apparent contradiction.
12. The authors recommend applying sunscreen before performing dPDT. Please clarify whether this is a correct strategy or is there an inaccuracy in the text? To which areas should it be applied?
13. It is recommended to change the format of the Figure 1 presentation.
14. In Table 1., references must be included.
Author Response
C1: The text of the article does not contain enough references to the literature to support the statements made. The tables are completely devoid of references.
A: We have added the appropriate references to the tables. We have also thoroughly rechecked the entire manuscript to ensure that all statements are supported by the appropriate references.
C2: The text of the article contains many unsubstantiated conclusions that require the inclusion of references.
A: As stated above, we have thoroughly rechecked the entire manuscript to ensure that all conclusions are supported by the appropriate references.
C3: In the Methodology section, the authors indicate that the review included only studies published in Greek and English. However, the review does not contain references to articles in Greek.
A: Thank you for your comment. Our literature search did not identify any studies in Greek that were eligible for inclusion in our review.
C4: Please describe in more detail the side effects associated with the use of both cPDT and dPDT? What are their main manifestations and how often do they occur?
A: We included two tables (Table 3 & Table 4) that summarize the side effects of cPDT and dPDT and their frequency.
C5: What is the underlying mechanism, membrane damage or mitochondrial dysfunction leads to apoptosis? Explain what is meant by oxidative stress.
A: We revised the relevant text to ensure clarity.
C6: What is the dynamics of photosensitizer accumulation from the moment of application of ointment with PS to the formation of a sufficient amount of PpIX in actinic keratosis cells for effective treatment?
A: Additional details on photosensitizer accumulation have been included in the 'Review of the dPDT Procedure' section.
C7: The text of the article does not present an analysis of modern methods for monitoring the accumulation and photobleaching of PS in tissues.
A: We have revised the text to include information regarding modern methods for monitoring photosensitizer accumulation and photobleaching in tissues.
C8: How many dPDT sessions are needed to achieve full therapeutic effect?
A: We have noted in the text that AK is a chronic condition requiring close follow-up and repeated treatment cycles based on disease severity (e.g., number of lesions, presence of hyperkeratotic AKs). However, we did not find specific data in the literature regarding the exact number of therapy cycles a patient may need. We believe that close patient follow-up will ensure proper AK retreatment if necessary.
C9: What is the role of Langerhans cells in the dPDT Actinic Keratosis process? Do they influence the effectiveness of therapy or the immune response?
A: We have revised the 'Review of the dPDT Procedure' section to include information on the role of the immune system in the PDT treatment process.
C10: How painless is dPDT? The authors mention that dPDT is virtually painless. However, there are enough studies that mention the occurrence of pain during 5-ALA dPDT.
A: We agree with the reviewer that our sentence can be misunderstood. Studies indicate that dPDT can cause pain, though it is significantly less than in cPDT. To avoid confusion, we have removed the term ‘virtually painless’ from the ‘Review of dPDT Procedure’ section and replaced it with: ‘The gradual activation of the photosensitizing agent during daylight exposure is likely the main reason why this procedure causes significantly less pain compared to conventional PDT (cPDT).’"
C11: The authors mention that patients should go out into daylight within 30 minutes of applying PS. Please clarify whether it is recommended to start exposure to daylight immediately after applying the ointment or whether it is necessary to wait a certain amount of time. This interesting point makes sense to expand on in order to remove the apparent contradiction.
A: The current guidelines recommend starting daylight (or artificial daylight) exposure within 30 minutes of photosensitizer application. We have revised the text to improve clarity.
C12: The authors recommend applying sunscreen before performing dPDT. Please clarify whether this is a correct strategy or is there an inaccuracy in the text? To which areas should it be applied?
A: We revised the relevant section (i.e. Review of the dPDT procedure) to increase clarity.
C13: It is recommended to change the format of the Figure 1 presentation.
A: We modified the format of Figure 1 and added text in the Discussion section to provide a detailed explanation of the proposed practical algorithm depicted in the figure.
C14: In Table 1.,references must be included.
A: We have included the appropriate citations in Table 1, as also suggested by reviewer 2